# What is the appropriate timing for advance care planning according to patients and their relatives? A scoping review

Carolien Burghout[1,2,3]*, Sascha R. Bolt[3], Lenny M. W. Nahar-van Venrooij[2,3], Tineke J. Smilde[1,4], Carin C. D. van der Rijt[5], Eveline J. M. Wouters[3,6]

1 Department of hemato-oncology, Jeroen Bosch Hospital, 's-Hertogenbosch, the Netherlands, 2 Jeroen Bosch Academy Research, Jeroen Bosch Hospital, 's-Hertogenbosch, the Netherlands, 3 Tranzo, Tilburg School of Social and Behavioral Sciences, Tilburg University, Tilburg, the Netherlands, 4 National Health Care Institute, Diemen, The Netherlands, 5 Department of Medical Oncology, Erasmus Medical Center Cancer Institute, Rotterdam, the Netherlands, 6 Fontys University of Applied Science, School of Allied Health Professions, Eindhoven, the Netherlands

☒ These authors contributed equally to this work.
* c.burghout@jbz.nl

## Abstract

### Background

Advance care planning (ACP) is often initiated in the last phase of life. However, patients and relatives may need earlier conversations about their care preferences in the disease trajectory.

### Objective

To synthesize empirical research on the appropriate timing of ACP from the perspective of patients and relatives. Additionally, we investigated facilitators, challenges, and triggers related to that timing.

### Design and method

A scoping review was conducted using PubMed and CINAHL databases (up to April 2023; updated June 2025). Eligible studies focused on ACP timing from the perspectives of patients with cancer, heart or lung disease, and relatives. Two researchers independently screened papers and extracted data. Extracted data were clustered using thematic analysis.

**Results:** In total, 29 papers were included. Both patients' and relatives' perspectives regarding the appropriate timing of initiating ACP varied widely and encompassed the entire continuum from healthy state, through illness, until the end of life. Timing related facilitating factors for initiating ACP were: clear information about prognosis, readiness, ACP as part of standard care, and clarity about who initiates ACP. Timing challenges were categorized as patient- (individual needs, coping, mutual protection), illness- (prognosis or illness related uncertainty) and professional-related (reluctance,

**Data availability statement:** All relevant data are within the paper and its Supporting Information files.

**Funding:** The author(s) received no specific funding for this work.

**Competing interests:** The authors have declared that no competing interests exist.

time constraints). Additionally, patient-related (age, experiences in life) and trigger points in illness were identified.

## Conclusion

The wide range of perspectives regarding appropriate timing of ACP requires a personalized approach. Clinical triggers may provide guidance, though they are not universally applicable. Professionals should timely and regularly explore patients' preferences and readiness for initiating ACP conversations.

---

## Introduction

Advance care planning (ACP) generally refers to a process of structured communication in which patients explore and articulate their values, preferences, and goals for future medical care [1,2]. Although ACP is operationalized differently across studies, it typically involves high-quality conversations that help patients reflect on what matters most to them when facing treatment decisions. Ideally, ACP involves collaboration between clinicians, patients, and their loved ones to consider how care should be approached if, or when the patient's health deteriorates. Effective ACP communication with patients may contribute to better quality of life and care, lower rates of in-hospital death, and less aggressive medical treatment at the end of life [3–11]. Despite these benefits, studies show that ACP conversations are often initiated late in disease trajectories [12–16] or not initiated at all [17]. This may result in end-of-life care that does not align with the patients' preferences. Patients may then, for example, experience unwanted care transitions at the end of life [18] and they may be unable to die at their preferred place [18–20].

In theory, ACP could be discussed regardless of a person's age or state of health [1,2,21] or as soon as a person is confronted with a (chronic) illness or frailty [22] as ACP discussions start with exploring what really matters for a person during health and disease. However, there may be varying ideas on the appropriate timing of initiating ACP conversations. Previous research on the initiation of ACP primarily focused on the perspectives of professionals [ 22–25]. To date, the perspectives of patients and their relatives on the appropriate timing of ACP conversations remain unclear, and a comprehensive synthesis is lacking. Therefore, a scoping review was conducted to explore what is known about the appropriate moment of ACP conversations from the perspective of adult patients and/or relatives. The review focused on patients with at least one of the most prevalent life-shortening diseases: cancer, heart- or lung diseases [26,27]. In addition, we examined existing knowledge regarding the facilitators, challenges, and triggers related to the timing of these ACP conversations.

## Materials and methods

A scoping review was performed to synthesize findings from empirical research on the appropriate timing of ACP from the perspective of patients and relatives. The protocol was registered at the Open Science Framework on March 8th, 2023 (https://osf.io/mxf5k/) (S1 File Protocol).

The checklist Preferred Reporting Items for Systematic Reviews and Meta-Analyses for Scoping Reviews (PRISMA-ScR) [28] was used to report the review (S2 File Checklist).

## Information sources and search strategy

The literature search was conducted in two databases (PubMed and CINAHL) selected for their comprehensive coverage of original studies in the fields of medicine, nursing, and allied health, relevant to ACP for patients with cancer, lung, or heart diseases.

PubMed and CINAHL were searched to identify potentially relevant studies up to April 7th 2023. An updated search was performed until May 31th 2025 in both databases. A comprehensive search strategy was developed in conjunction with an experienced librarian at the Jeroen Bosch Hospital and was subsequently refined with the authors. The quality of the search strategy was tested by ensuring that a key set of relevant studies was retrieved. The search strategy was formulated using terms equivalent or relevant to the topic of ACP (e.g., end-of-life conversations) and using diverse terms for 'timing'. These were combined with terms representing adults with the selected life-shortening illnesses, cancer, heart- and lung diseases. S3 File presents the final search strategy. The studies retrieved were exported into EndNote and duplicates were removed.

## Eligibility criteria

Articles were included if they addressed the timing of ACP, explicitly or implicitly, based on the study's objectives or methods, and focused on adult patients with one of the three most prevalent life-shortening diseases (cancer, heart or lung disease) and/or their relatives [26,27].

Studies also had to clearly describe the preferences of patients or relatives regarding end-of-life care. Studies that focused on specific ACP topics such as living wills, the naming of a patient representative, do-not-resuscitate or do-not-intubate orders and dialysis decision-making were only considered if they also mentioned end-of-life preferences. Mixed-population studies, containing both eligible and ineligible groups, were included only if outcomes for the eligible population were reported separately.

## Study selection

The online software Covidence (www.covidence.org) [28] was used to support the screening and selection process of papers according to international guidelines [29,30]. Initially, titles and abstracts were screened to exclude articles that were irrelevant or beyond the scope of this review in accordance with the study protocol (S1 File protocol). Screening was primarily performed by one reviewer with partial independent verification, an approach considered methodologically acceptable while enhancing feasibility [31]. CB screened all titles and abstracts, while EW, LN, and SB independently screened a subset until consensus was reached. After the initial screening phase, articles considered suitable for full-text review were further assessed. CB screened all articles in the full-text phase, and EW, LN, or SB independently screened subsets. In both phases, discrepancies were resolved through discussion. Reasons for the exclusion of full-text papers were reported.

## Data extraction

A data-charting form was developed by CB and refined with the authors. For each study, descriptive data were extracted including author, title, year of publication, country of origin, study aims, study design, type of data collection, setting, study population, eligibility criteria, number of participants, results about (in)appropriate timing of ACP from the perspective of patients and/or their relatives, and relevant facilitators, challenges, and triggers related to the timing of ACP. Data extraction was performed independently by CB, with SB extracting a subset. Discrepancies were resolved through discussion until consensus was reached.

## Data analysis

This scoping review explored appropriate timing for initiating ACP conversations about end-of-life care preferences from the perspective of patients with diverse life-shortening diseases and their relatives. The review did not mean to address differences in perspectives between disease groups. Rather, it aimed to derive a broader image on ACP timing from the perspective of a combined patient population.

Relevant findings from quantitative studies were transformed into textual descriptions to enable consistent thematic coding and integration with qualitative data [32].

A thematic analysis was performed on the extracted content, using mixed inductive and deductive coding [33]. CB performed line-by-line open coding of all included studies, while SB independently coded a subset. Differences in coding were discussed until consensus was reached. In cases where CB coded studies independently, SB was consulted to resolve uncertainties related to the interpretation or coding of specific text fragments. The codes related to appropriate timing were gathered and clustered by CB and SB into timeframes that were mentioned as (in)appropriate from different perspectives. The codes related to facilitators, challenges, and triggers about ACP timing were clustered under each topic. During the clustering process, several reflection sessions were held with all authors to reach a consensus about the clustering and defining of the subthemes.

ATLAS.ti software program, version 24.2.0.32043 [34] supported the data analysis.

## Results

### Study selection

The initial search in PubMed and CINAHL in April 2023 retrieved 9,448 records, of which 581 were duplicates. The remaining 8,867 records were screened for eligibility based on title and abstract by CB, with a subset of 854 (9.6%) independently double-screened by EW, LN, or SB. Following the double-screening phase, discrepancies were resolved, after which the remaining title and abstract screening was completed by CB. This resulted in 84 articles selected for full-text review. CB assessed all full texts, with a subset of 15 (17.9%) independently double-reviewed by EW, LN, or SB, resulting in 25 articles meeting the inclusion criteria.

The updated search up to May 2025 retrieved 2,816 additional records, including 518 duplicates. CB screened all the remaining 2,298 records, selecting 22 for full-text review, of which four met the inclusion criteria. Any uncertainties regarding study inclusion or exclusion were resolved through consultation with LN until consensus was achieved. Together, both searches yielded 29 papers that were fit for inclusion. Fig 1 presents the PRISMA flow chart of the study selection process.

### Characteristics of included studies

Table 1 summarizes the characteristics of the included articles. Studies were conducted in fourteen different countries, most often Western countries (25/29). Most studies were from the USA (n = 7), Canada (n = 5), and United Kingdom (n = 4). The study designs were qualitative (n = 19), quantitative (n = 5), or mixed methods (n = 5). Among the included studies, interviews (n = 20), surveys (n = 7), and focus groups (n = 5) were the most common design types, with some studies combining multiple methods. Seventeen studies described the sole perspective of patients (n = 14) or relatives (n = 3), while twelve studies described a combination of these perspectives. Most studies included patients or relatives related to cancer (n = 21), followed by pulmonary disease (n = 4), and heart disease (n = 3). One study included a population with cancer or lung disease and described the results separately per disease. Most of the studies were conducted in a hospital (n = 22). Other settings were a mixed setting (n = 5) (i.e., hospital and hospice, general practice and hospital). Two studies were conducted online whereby participants were recruited via the internet.

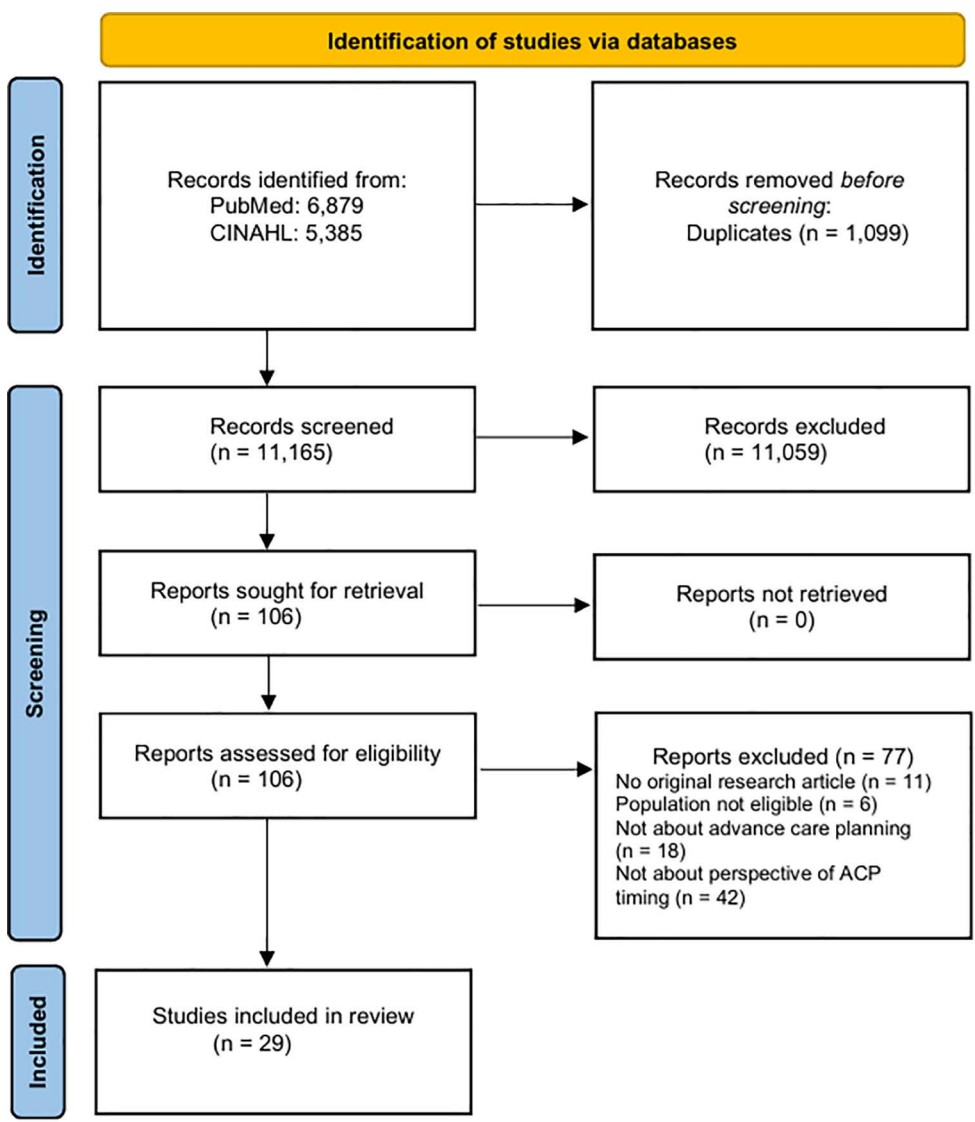

**Fig 1. PRISMA flow chart of the study selection process.**

### Timing of ACP in the disease course

There was a considerable variation in perspectives on the (in)appropriate timing of ACP conversations in both patients and relatives, covering the entire path from healthy state, followed by illness, until the end of life. The thematic analysis identified the following moments: during a healthy state, early/earlier in the disease process, before and after deterioration, before, during and after treatment, and at the end-stage of the disease (For more detail see Table 2 and S4 File Code tree). Some of the moments were also regularly mentioned in the opposite direction, indicating inappropriate timing. These timeframes are not mutually exclusive and may overlap. For instance, 'early in the disease course' may coincide with the period 'before treatment'.

Across cancer, heart and lung disease, many patients and relatives considered the moment soon after being diagnosed or early in the disease process as appropriate for initiating ACP [35,36,43–50], when the patient has little to no disease

**Table 1. Overview of the included studies.**

| Author | Year of publication | Country of origin | Aim of the study | Study design | Type of data collection | Setting | Partici-pants | Number of participants |
|---|---|---|---|---|---|---|---|---|
| **Articles Oncology** | | | | | | | | |
| **Perspective of patients** | | | | | | | | |
| Pendle-ton D. | 2025 | USA | To characterize patient beliefs and experiences pertaining to ACP discussions, and to explore factors that may be associated with patient experiences. | Quan-titative research | Survey | Hospital | Patients with cancer | 100 |
| Trevi-zan F.B. | 2023 | Brazil | To identify the patients who are most likely to participate in discus-sions about palliative care (PC) and advance care planning (ACP), and to determine their preferred timing and approach of discussion. | Mixed method | Survey, interview | Hospital | Patients with cancer | 115 |
| Zhu T. | 2023 | The Nether-lands, Belgium, Slovenia, Italy, Denmark, and the United Kingdom | To explore the experiences of patients with advanced cancer regarding the timing of ACP. | Quan-titative research | Survey | Hospital | Patients with cancer | 288 |
| Kubi B. | 2020 | USA | To determine oncology patients' preferences surrounding ACP with a focus on the choice of which health care providers to have the conversation with and the timing of conversations. | Mixed method | Survey, other (medical record review) | Hospital | Patients with cancer | 200 |
| Seifart C. | 2020 | Germany | To investigate gender differences concerning the content, the desired time point, and the mode of initiation of EOL conversations in cancer patients. | Qual-itative research | Interview | Mixed setting (hospital, rehabilitation clinic) | Patients with cancer | 186 |
| Flied-ner M. | 2019 | Switzerland | To explore advanced cancer patients' experiences with a struc-tured early palliative care interven-tion, its acceptability and impact on the patients' life including influencing factors. | Qual-itative research | Interview | Hospital | Patients with cancer | 20 |
| Ólafs-dóttir K.L. | 2018 | Iceland | To explore the experiences of patients newly diagnosed with advanced lung cancer and their family members of engaging in a person-centered and structured ACP discussion. | Qual-itative research | Interview | Hospital | Patients with cancer | 7 |
| Sher-man A.C. | 2018 | USA | To deepen our understanding of preparedness for EOL care, from the perspective of patients themselves. | Qual-itative research | Interview | Hospital | Patients with cancer | 13 |
| Michael N. | 2013 | Australia | To further an appreciation of how cancer patients consider ACP so that patients' responses could inform the development of an ACP program in an Australian Cancer Centre. | Mixed method | Interview, other (Vignettes) | Hospital | Patients with cancer | 18 |

*(Continued)*

**Table 1.** (Continued)

| Author | Year of publication | Country of origin | Aim of the study | Study design | Type of data collection | Setting | Participants | Number of participants |
|---|---|---|---|---|---|---|---|---|
| **Barnes K.A.** | 2011 | United Kingdom | To explore the suitability, nature, and efficacy of ACP discussions from the perspectives of oncology patients and carers. | Qualitative research | Other (audio-taped consultations) | Mixed setting (hospital, hospice) | Patients with cancer | 40 |
| **Perspective of relatives** | | | | | | | | |
| **Hayashi Y.** | 2022 | Japan | To determine the (1) status of end-of-life discussions in general wards, (2) associated factors, and (3) impact of end-of-life discussions on outcomes. | Quantitative research | Survey | Hospital | Relatives of patients with cancer | 119 |
| **Perspective of patients and relatives** | | | | | | | | |
| **Sripaew S.** | 2024 | Thailand | To explore the perceptions of older cancer patients and their families towards ACP engagement | Qualitative research | Interview | Hospital | Patients with cancer, and relatives | 32 (16 patients, 16 relatives) |
| **Canny A.** | 2022 | Scotland | To assess the feasibility and acceptability to patients, carers, and GPs of a primary care ACP intervention. | Mixed method | Interview, survey | Mixed setting (general practice, hospital) | Patients with cancer and relatives | 23 patients, unknown amount relatives |
| **Ikander T.** | 2022 | Denmark | To investigate current nursing practice related to end-of-life discussions with incurable lung cancer patients and their family caregivers from the perspectives of patients, family caregivers, and nurses in an oncology outpatient clinic. | Mixed method | Focus group, interview, observation | Hospital | Patients with cancer and relatives | 28 (9 patients, 8 relatives) |
| **Sato A.** | 2022 | Japan | To assess the need for a question prompt list (QPL) that encourages end-of-life discussions between patients with advanced cancer and their physicians. | Qualitative research | Interview, focus group | Hospital | Patients with cancer, and relatives | 18 (5 patients, 7 relatives) |
| **Rodi H.** | 2021 | Australia | To provide more clarity regarding ACP processes in Australian oncology settings to enable widespread uptake across the cancer system. | Quantitative research | Survey | Other (national online study) | Patients with cancer and relatives | 705 (440 patients, 265 relatives) |
| **Cutshall N.R.** | 2020 | USA | To review a social media tweet chat to understand barriers to ACP experienced by brain tumor stakeholders. | Qualitative research | Other (tweet analyses) | Other (Forum on internet) | Patients with cancer and relatives | 52 tweet chat participants (21 patients, 5 caregivers) |
| **Fritz L.** | 2020 | The Netherlands | To develop an ACP program specifically for glioblastoma patients and assess the preferred content, the best time to introduce such a program in the disease trajectory, and possible barriers and facilitators for participation and implementation. | Qualitative research | Interview, focus group | Hospital | Patients with cancer and relatives | 23 (8 patient-proxy dyads, 5 proxies of deceased patients) |

*(Continued)*

**Table 1.** (Continued)

| Author | Year of publication | Country of origin | Aim of the study | Study design | Type of data collection | Setting | Partici-pants | Number of participants |
|---|---|---|---|---|---|---|---|---|
| Toguri J.T. | 2020 | Canada | To: 1) explore patients' and families' understanding, experience and reflections on ACP, as well as what they need from their physicians during the process; 2) explore physicians' views of ACP, including their experiences with initiating ACP and views on ACP training. | Qualitative research | Interview | Hospital | Patients with cancer, relatives | 18 (4 patients, 4 relatives) |
| Booker R. | 2018 | Canada | To understand the barriers to and facilitators of ACP in hematopoietic stem cell transplantation. | Qualitative research | Interview | Hospital | Patients with cancer and relatives | 13 (6 patients, 5 relatives) |
| Barnes K. | 2007 | United Kingdom | To explore the acceptability of an interview schedule, designed to encourage conversations regarding future care; and to explore the suitability of such discussions and inquire about their possible timing, nature and impact. | Qualitative research | Focus group | Mixed setting (hospital, hospice) | Patients with cancer and relatives | 22 (18 patients, 4 relatives) |
| **Articles lung diseases** | | | | | | | | |
| **Perspective of patients** | | | | | | | | |
| Tavares N. | 2020 | United Kingdom | To understand the preferences of patients with chronic obstructive pulmonary disease for discussions about palliative and advance care planning with clinicians. | Qualitative research | Interview | Mixed setting (primary care, respiratory outpatient clinics, research department) | Patients with lung disease | 33 |
| Nguyen M. | 2013 | Canada | To explore the perceived ACP needs of people with COPD at different illness severities and how these are met by a DVD discussing ACP. | Qualitative research | Interview | Hospital | Patients with lung disease | 12 |
| **Perspective of relatives** | | | | | | | | |
| Pooler C. | 2018 | Canada | To explore bereaved caregivers' experiences of IPF patients' end-of-life care with the palliative approach initiated at the first visit to the clinic. | Qualitative research | Interview | Hospital | Relatives in lung disease | 8 |
| **Perspective of patients and relatives** | | | | | | | | |
| Kalluri M. | 2022 | Canada | To explore perspectives of IPF patients, family caregivers, and healthcare professionals on ACP-related experiences to understand and inform an ACP framework to guide clinicians and facilitate early, meaningful conversations. | Qualitative research | Interview | Hospital | Patients with lung-disease and relatives | 20 (5 patients, 5 relatives) |
| **Articles heart diseases** | | | | | | | | |
| **Perspective of patients** | | | | | | | | |
| Dzou T. | 2022 | USA | To identify and describe MCS (mechanical circulatory support) individuals' perceptions of opportunities and challenges for ongoing ACP communication. | Qualitative research | Interview | Hospital | Patients with heart disease | 24 |

*(Continued)*

**Table 1.** (Continued)

| Author | Year of publication | Country of origin | Aim of the study | Study design | Type of data collection | Setting | Partici-pants | Number of participants |
|---|---|---|---|---|---|---|---|---|
| **Perspective of relatives** | | | | | | | | |
| Chuzi S. | 2021 | USA | To understand bereaved caregiver and clinician perceptions of end-of-life discussions and care for patients with DT LVADs. | Qual-itative research | Interview | Hospital | Rela-tives of patients with heart disease | 7 |
| **Perspective of patients and relatives** | | | | | | | | |
| Metzger M. | 2016 | USA | To increase our understanding of patients' and surrogates' experience of engaging in ACP discussions, specifically how and why these dis-cussions may benefit patients with LVADs and their families. | Qual-itative research | Interview | Hospital | Patients with heart diseases and relatives | 28 (14 patients/sur-rogate pairs) |
| **Articles with a mixed patient population** | | | | | | | | |
| Hjorth N.E. | 2018 | Norway | To explore pulmonary patients' needs and preferences regarding ACP in order to prepare for the introduction of ACP in Norwegian hospitals | Qual-itative research | Focus group | Hospital | Patients with cancer or lung disease | 13 |

burden [46,39,40,51]. However, a few patients considered discussing ACP early in the disease process as too early [22,44,37,52,53], especially when a patient is emotionally burdened [54] or when the diagnosis is still doubtful [44]. Start-ing ACP during a healthy state was mentioned as an appropriate moment by a minority of patients [35,39,55].

Patients and relatives often considered the period before disease deterioration as the appropriate time to initiate ACP, as then the patient is not in crisis [46,47,35,39,37,54,56] and does not have cognitive impairments [51,54,57]. This was mentioned by all patients and in all disease groups. Some studies, however, reported that patients preferred ACP after deterioration, for instance when symptoms are worsening [37,55,38].

Perspectives on when to initiate ACP varied across the treatment pathway in patients with cancer and heart disease: before, during, or after treatment [45,40,39,56,58,41]. For example, patients with heart disease held conflicting views about discussing ACP in relation to cardiac device placement, with some preferring ACP conversations afterward, while others mentioned that moment as inappropriate [56].

Some patients and relatives mentioned that having an advanced disease may be an appropriate moment to start ACP, such as when curative treatment options are no longer available [47,49], upon transition to palliative care [58], or in the most severe stage of COPD [46]. However, one patient with cancer mentioned that initiating ACP during palliative care while still feeling well was inappropriate [40]. Relatives of patients with cancer identified several appropriate moments to initiate ACP, including three months before death [42], when treatment is no longer beneficial [44], or when the patient has entered the terminal phase [42,59]. One patient with cancer stated that ACP is a dynamic process and that it should take place at three key moments: when the patient is still in a healthy state, when symptoms first appear, and when symptoms worsen [55].

### Facilitators related to the timing of ACP

Facilitating factors for the timing of ACP were clustered into the following subthemes: clear information about prognosis and expectations, acceptance/readiness, ACP as part of standard care, and clarity about who initiates ACP (Table 3 and

Table 2. Appropriateness of ACP timing according to patients and relatives.

| Subtheme | Timing | Stated by | | | | | | References | Illustrating quotes or text passages | Type* |
|---|---|---|---|---|---|---|---|---|---|---|
| | | Patient | | | Relative | | | | | |
| | | Ca | HD | LD | Ca | HD | LD | | | |
| During healthy state | Appropriate | ✓ | | | | | | 44, 46, 52 | "(..) with one individual expressing that ACP should be discussed prior to any serious diagnosis" [35]. | Patient's quote |
| Early/earlier in disease process | Appropriate | ✓ | ✓ | ✓ | ✓ | | ✓ | 35-47, | "To me, I think that is something that (ACP) should be at the beginning" [36]. | Patient's quote |
| | Inappropriate | ✓ | ✓ | ✓ | ✓ | | | 22, 36, 49-51, 60 | | |
| Before deterioration | Appropriate | ✓ | ✓ | ✓ | ✓ | ✓ | | 38, 40, 44, 46, 47, 49, 51, 53, 54 | "Well, anytime (now). Whenever it's convenient with anyone like... You know? Because at the moment I'm in good health. So, I would really like to, I suppose, talk to someone" [37]. | Patient's quote |
| After deterioration | Appropriate | ✓ | | ✓ | | | | 49, 52, 55 | "If...Dr [x] said to me, "look...it's flaring up again." and if it was, then I think I'd say, "well, now let's plan" [38]. | Patient's quote |
| Before treatment | Appropriate | ✓ | ✓ | | ✓ | ✓ | | 46, 51, 53, 56, 57 | "For patients who preferred to have discussions during the course of their cancer therapy (6.5%), they typically preferred to have them before starting therapy" [39]. | Result (author) |
| During treatment | Appropriate | ✓ | | | ✓ | | | 45, 56 | "Most patients and family caregivers wanted to talk about the future at different times throughout treatment" [40]. | Result (author) |
| After treatment | Appropriate | ✓ | ✓ | | ✓ | | | 37, 53, 57 | "In total, the majority of the interviewed cancer patients would like to talk about any topic when their disease is getting worse (58%); 27.5% prefer to talk at the end of therapy" [41]. | Result (author) |
| | Inappropriate | | ✓ | | | | | 51, 53 | | |
| End stage (palliative/terminal phase) | Appropriate | ✓ | | ✓ | ✓ | | | 37, 38, 42, 56, 59 | "Whereas some wanted to discuss their EoL as soon as possible, others wanted to delay it until the very end" [42]. | Result (author) |
| | Inappropriate | ✓ | | | | | | 45 | | |

ACP: advance care planning, Ca: cancer; HD: heart disease; LD: lung disease; EoL: end-of-life

* Some excerpts are direct quotes from patients or relatives, while others are author interpretations.

S4 File Code tree). These facilitators were especially mentioned by patients with cancer or lung diseases. Patients and relatives mentioned that having clear information about the prognosis and what to expect in the future helps to initiate ACP early. Ideally, this information is given at an early stage [36,35]. A few patients mentioned that professionals may start ACP when patients show readiness [40,56], for example when patients initiate the conversation themselves [49,38]. One patient emphasized the importance of receiving prior notice, allowing them to prepare and to make sure the right people can join the conversation [46]. A close care relationship was mentioned as a prerequisite for initiating ACP by one patient [44]. In a few studies, patients stated that initiating ACP would be easier if integrated into annual check-ups or standard care [39,37] or during the support service in such a natural way that participants felt it had not been deliberately planned [47]. Some patients emphasized the need for clarity about who initiates ACP. They reported waiting for their professional to start the conversation, regarding it primarily as the professional's responsibility [52,55,38].

## Challenges related to the timing of ACP

Diverse challenges related to the timing of ACP were mentioned by patients and relatives, and these included the subthemes patient-related (individual needs, coping, mutual protection), illness/treatment-related (prognosis or illness uncertainty), and professional-related challenges (reluctance, time constraints) (Table 3 and S4 File Code tree). From all

Table 3. Facilitators, challenges and triggers related to the appropriate timing of ACP according to patients and relatives.

| Subtheme | Stated by | | | | | | Illustrating quotes or text passages | Type* | References |
|---|---|---|---|---|---|---|---|---|---|
| | Patient | | | Relative | | | | | |
| | Ca | HD | LD | Ca | HD | LD | | | |
| **Facilitators related to the appropriate timing of ACP** | | | | | | | | | |
| Having clear information about prognosis and expectations | ✓ | | ✓ | ✓ | | ✓ | "I want to know what to expect, what's coming down the line. To me, I think that is something that should be at the beginning" [36]. | Patient's quote | 39, 44 |
| Acceptance/ readiness | ✓ | ✓ | ✓ | | | | "You need to fit it in with them when they are ready" [56]. | Patient's quote | 36, 38, 42, 45, 49, 53, 55 |
| ACP as part of standard care | ✓ | | ✓ | | | | "Participant comments revealed a wish to incorporate ACP discussions into the annual physical checkup" [39]. | Result (author) | 40, 46, 49 |
| Clarity about who initiates ACP | ✓ | | | | | | "A small number of patients said they would take the lead from health professionals on when to have conversations" [38]. | Result (author) | 48, 52, 55 |
| **Challenges related to the appropriate timing of ACP** | | | | | | | | | |
| **Patient-related** | | | | | | | | | |
| Individual needs | ✓ | ✓ | ✓ | ✓ | ✓ | | "The participants varied widely in their opinions of the best time to broach ACP. Whereas some wanted to discuss their EoL as soon as possible, others wanted to delay it until the very end. Overall, it seems that most of the participants simply requested the medical team be sensitive to their needs on an individual basis" [59]. | Result (author) | 22, 35-38, 40, 49, 53, 55, 59, 60 |
| Coping | ✓ | ✓ | ✓ | ✓ | | | "Several patients explained that they preferred to concentrate on positive outcomes and felt that ACP was an indication or concession that treatment would be unsuccessful: 'I think that's [ACP] an acceptance that it is going to go bad' [60]. | Result (author)/ patient's quote | 22, 40, 41, 45, 48, 49, 55 60, 61 |
| Mutual emotional protection | ✓ | ✓ | | ✓ | ✓ | | "Because the subject of death and dying was uncomfortable, the participants seemed hesitant about initiating the discussion with their family, to protect both them and their family. One patient wondered if people were ever ready to discuss these matters: "Well, are you ever ready? [silence] I hope that I will live a little bit longer, hopefully" [47]. | Result (author)/ patient's quote | 40, 45, 48, 53 |
| **Illness/ treatment-related** | | | | | | | | | |
| Prognosis or illness uncertainty | ✓ | | | | ✓ | | "I could see him deteriorating. His energy level was deteriorating. But did I suspect that he was doing to die? No. So was I prepared? No." [61]. | Relative's quote | 60, 62 |
| **Professional-related** | | | | | | | | | |
| Reluctance | ✓ | | | | | | "Over a third said their doctors were reluctant to introduce such topics. They always try to be positive.... upbeat... So, he's not going...to say, "What happens if it goes wrong?" He doesn't want to discuss it" [38]. | Result (author)/ patient's quote | 55 |
| Time constraints | ✓ | | | ✓ | | | "The doctors...are very busy...so I have not talked to them, because it is probably quite a lengthy subject" [38]. | Patient's quote | 54, 55 |
| **Triggers related to the appropriate timing of ACP** | | | | | | | | | |
| **Patient-related** | | | | | | | | | |
| Age | ✓ | | | | | | "Many patients commented on the possible benefit of establishing guidelines for beginning ACP discussions by a certain age, similar to other screening interventions" [39]. | Result (author) | 46 |
| Experiences in life | | ✓ | ✓ | | | | "When I was hospitalized here, there was a man next to me. He may have passed away by now, but let's just say that wasn't the way that I would like to end my life. I plan on taking every precaution necessary to never end up that way" [59]. | Patient's quote | 51, 59 |

*(Continued)*

**Table 3.** (Continued)

| Facilitators related to the appropriate timing of ACP | | | | | | | | |
|---|---|---|---|---|---|---|---|---|
| Subtheme | Stated by | | | | | | Illustrating quotes or text passages | Type* | References |
| | Patient | | | Relative | | | | | |
| | Ca | HD | LD | Ca | HD | LD | | | |
| Trigger points in illness | | | | | | | | | |
| Turning points | ✓ | ✓ | ✓ | ✓ | | | "The participants named key turning points that often were consistent with major medical changes. Examples of such turning points were an infection triggering change, a new metastasis, increasing pain, increasing dyspnea, loss of a function, decline in their general condition and stopping chemotherapy" [46]. | Result (author) | 22, 38, 44, 49, 51, 52, 57 |

ACP: advance care planning, Ca: cancer; HD: heart disease; LD: lung disease; EoL: end-of-life

* Some excerpts are direct quotes from patients or relatives, while others are author interpretations.

perspectives and for all disease groups, it was emphasized that the appropriate timing of ACP depends on the person and should be tailored to the individual patient [22,43–46, 37,56,38,59,62]. Finding the right time to initiate ACP was also depend on how patients and their relatives cope with the illness. Focusing on curation might lead to patients avoiding ACP conversations [62,60]. In a few studies, patients with cancer and lung diseases stated that talking about death was difficult or uncomfortable for them [47,48,52,37]. Patients and relatives reported that the timing of ACP is experienced differently [56], with the initiation of ACP being perceived either as signaling a poor outcome [60] or as confirming the seriousness of the illness [47]. Family discussions are further hindered by efforts to protect each other emotionally. For example, patients and relatives mentioned hesitancy to discuss ACP with each other because they did not want to cause pain or distress in the other [47,40]. Several patients with cancer shared that having cancer shifted their focus to the present. While open to certain conversations, they valued retaining control by choosing when to continue, pause, or end such conversations [22].

In addition to patient-related challenges, illness/treatment related challenges were also mentioned as influencing the timing of ACP. Relatives of patients with heart diseases mentioned that an unpredictable disease trajectory may hinder the initiation of ACP [61]. A patient with cancer mentioned that a perceived good health and prognosis may delay the initiation of ACP [62].

Lastly, patients described professional-related challenges related to reluctance with regard to the initiation of ACP. Patients reported that ACP conversations were often postponed or avoided because professionals appeared reluctant to introduce discussions about future health deterioration or end-of-life care [38]. In addition, patients and relatives experienced that time taken to engage ACP conversations was insufficient [57,38].

## Triggers related to the timing of ACP

Triggers were defined as moments or changes in a patient's situation that may prompt the start of ACP. We found patient-related triggers (age, experiences in life) and trigger points of illness (turning points) (Table 3 and S4 File Code tree). From the perspective of patients, reaching a certain age and experiences in life (i.e., witnessing others at the end of life, the death of a person close to the patient) were mentioned as patient-related triggers to start ACP [39,54,59]. Most commonly mentioned trigger points of illness by patients were: decline in general condition or loss of a function [46], treatment failure or recurrence of illness [22,46,41], a new metastasis [46], exacerbation of COPD [37], increasing dyspnea [46], more pain, or the occurrence of symptoms or complications [54,55]. In the study of Toguri et al. [35], relatives of cancer patients pointed out that any potential medical problem could indicate an appropriate moment to start ACP.

## Discussion

This scoping review examined the literature on appropriate timing for initiating ACP conversations about end-of-life care preferences, from the perspective of patients with a life-shortening disease and their relatives. We found a wide variation in what was considered an appropriate moment, encompassing the entire continuum from being in a healthy state, through illness, until the end of life.

The wide variation in the perceived appropriate timing of ACP conversations about end-of-life suggests that ACP timing is a highly individual matter. This underlines previous notions of the importance of understanding each patient's personal circumstances [63,64] and social background [65] when healthcare professionals want to initiate ACP conversations. Furthermore, cultural, national, and institutional factors may shape perceptions of the appropriate timing of ACP. Most included studies were conducted in Western countries, where patient autonomy is highly valued [66], which may facilitate openness to early ACP conversations. In contrast, in non-Western or family-centered cultures, decision-making may be more collective or deferential to medical authority, influencing both willingness to engage in ACP and perceptions of timing [67]. Similarly, countries with well-established ACP systems and training for care providers may support earlier and more structured discussions, whereas in countries without such frameworks ACP may occur later or structured in a less formal manner [68]. Individual patient characteristics, social background, as well as cultural, national, and institutional contexts are likely to contribute to the wide variation in perceived timing observed in this review, highlighting the importance of considering these multiple contexts when initiating ACP conversations. Patients may wish to discuss end-of-life preferences early, shortly after diagnosis or even before illness onset, indicating that the timing of ACP should not be tied exclusively to clinical milestones or disease progression. Miyashita et al. [69] showed that willingness of early ACP conversations is linked to factors like age, strong social support, and rejection of life-sustaining treatments. These findings call for a shift in practice, as healthcare professionals often still associate ACP solely with the palliative or terminal phase of the disease [61,70,71].

Given the wide variety of potentially appropriate moments for initiating ACP conversations mentioned by patients and their relatives, it is crucial for healthcare professionals to discern when they are open to starting these conversations, and equally important to recognize and respect when they are not (yet) ready. This requires sensitivity to patients' and relatives' fluctuating readiness, as observed in the ACTION study, where patients displayed both readiness and resistance during ACP conversations [72]. Education and training may support healthcare professionals in making more informed and sensitive decisions about when and how to initiate ACP, especially early in the disease course.

Our findings further suggest that timing is closely linked to perceptions of responsibility for initiation. Patients who view initiation as the professionals' responsibility are unlikely to take the first step themselves, making the professionals' proactive stance decisive for whether and when ACP occurs. In addition, professionals should be aware that their own values, norms, and personal views may consciously or unconsciously shape the timing and content of ACP conversations [67,73,74]. Targeted training that addresses both these relational dynamics and self-awareness is therefore recommended.

In our review, patients identified having clear information about their disease and prognosis as a key facilitator for initiating ACP conversations, which aligns with the findings of Shen et al. [75]. However, healthcare professionals often experience difficulties in prognostication, especially in non-cancer illnesses [76–78]. In conditions like COPD or heart failure, patients may be less aware of the terminal nature of their disease, which may contribute to professionals' reluctance to discuss prognosis and initiate ACP conversations [79,80]. Even in oncology, recent advances such as immunotherapies and other targeted therapies have added complexity to prognostication, making it more challenging to determine and communicate an accurate prognosis. Furthermore, studies show that clinicians tend to overestimate patients' disease outcomes [81–83], which may further delay the initiation of ACP conversations. Normalizing the addressing of ACP in a timely manner, regardless of a patient's prognosis, may help to overcome several challenges identified in this review, such as prognostic uncertainty, its association with severe illness, and concerns about causing distress.

The triggers related to timing identified in this review are often situated late in the disease trajectory, such as disease recurrence, treatment failure, or symptom progression. These late-stage triggers are also commonly used by healthcare professionals [43,84] to initiate ACP. While such triggers can be useful in recognizing moments to initiate or readdress ACP, acting only when these triggers occur reflects a reactive rather than a proactive approach. For some patients, these moments may come too late, as they might have preferred to discuss their preferences earlier in the disease trajectory. This again underscores the importance of healthcare professionals actively exploring, preferably at an early stage, which moments are most appropriate for initiating and readdressing ACP with each individual patient, based on their unique context and preferences.

## Limitations

The review may not be fully exhaustive, because inclusion required that abstracts explicitly or implicitly, through study objectives or methods, addressed timing. Studies with a broader ACP focus that would nonetheless offer relevant insights into timing and related facilitators, challenges, or triggers may have been inadvertently excluded.

The studies included in this review were predominantly conducted in Western countries, where autonomy is a highly valued principle in healthcare [66]. This cultural context may limit the generalizability of the findings with different cultural backgrounds and to cultural minorities. Prior research has emphasized the influence of ethnicity on ACP, as these factors shape perceptions of end-of-life care and the receptivity to such care [67]. However, the limited representation of studies from non-Western settings reflects a gap in the existing literature rather than a limitation of this review itself. Future research should therefore seek to address this gap by exploring perspectives on ACP in more diverse cultural contexts.

Most studies were conducted in the field of oncology, with only a few including patients or relatives with lung or heart disease. In addition, the perspective of relatives was generally underrepresented. Although it is plausible that the optimal timing of ACP may differ across conditions such as cancer, heart disease, or lung disease, the qualitative nature of the included results and the predominance of studies conducted in oncology, together with the underrepresentation of studies including relatives, did not allow for meaningful comparisons between disease groups or between patients and relatives. Rather than focusing on potential differences, this review aimed to synthesize the perspectives of patients and relatives across life-shortening illnesses in order to identify overarching patterns. Despite these limitations, the results across these various perspectives support the notion that the optimal timing for initiating ACP is highly individual among all of these groups.

## Conclusion

There is considerable variation in what is perceived as the appropriate timing for initiating ACP, covering the entire continuum from a healthy state to the end of life. Identifying the right moment is complex and calls for a tailored, personalized approach. While clinical or situational triggers may help guide timing, they are not universally applicable and may differ across patients and relatives. Normalizing the addressing of ACP in a timely manner, regardless of a patient's prognosis, may help overcome several challenges. Healthcare professionals should actively explore, preferably at an early stage, which timing is most appropriate for each patient, based on their unique context and preferences, and their relatives.

## Supporting information

**S1 File. Protocol.**
(DOCX)

**S2 File. PRISMA-ScR checklist.**
(DOCX)

**S3 File. Search string.**
(DOCX)

**S4 File. Code tree.**
(DOCX)

## Acknowledgments

The authors thank Kelly Cooijmans, PhD, Information specialist, Jeroen Bosch Academy Research, Jeroen Bosch Hospital for her contribution in creating the search string.

## Author contributions

**Conceptualization:** Carolien Burghout, Sascha R. Bolt, Lenny M.W. Nahar-van Venrooij, Tineke J. Smilde, Carin C.D. van der Rijt, Eveline J.M. Wouters.

**Data curation:** Carolien Burghout, Sascha R. Bolt, Lenny M.W. Nahar-van Venrooij, Tineke J. Smilde, Carin C.D. van der Rijt, Eveline J.M. Wouters.

**Formal analysis:** Carolien Burghout, Sascha R. Bolt, Lenny M.W. Nahar-van Venrooij.

**Investigation:** Carolien Burghout, Sascha R. Bolt, Lenny M.W. Nahar-van Venrooij.

**Methodology:** Carolien Burghout, Sascha R. Bolt, Lenny M.W. Nahar-van Venrooij, Tineke J. Smilde, Carin C.D. van der Rijt, Eveline J.M. Wouters.

**Project administration:** Carolien Burghout.

**Software:** Carolien Burghout, Sascha R. Bolt.

**Supervision:** Sascha R. Bolt, Lenny M.W. Nahar-van Venrooij, Tineke J. Smilde, Carin C.D. van der Rijt, Eveline J.M. Wouters.

**Validation:** Carolien Burghout, Sascha R. Bolt, Lenny M.W. Nahar-van Venrooij.

**Visualization:** Carolien Burghout.

**Writing – original draft:** Carolien Burghout.

**Writing – review & editing:** Sascha R. Bolt, Lenny M.W. Nahar-van Venrooij, Tineke J. Smilde, Carin C.D. van der Rijt, Eveline J.M. Wouters.

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
