## [Decision Letter · Decision Letter 0]

16 Dec 2025

PONE-D-25-49791What is the appropriate timing for advance care planning according to patients and their relatives? A scoping review .PLOS One?

Dear Dr. Burghout,

Thank you for submitting your manuscript to PLOS ONE. After careful consideration, we feel that it has merit but does not fully meet PLOS ONE’s publication criteria as it currently stands. Therefore, we invite you to submit a revised version of the manuscript that addresses the points raised during the review process.

We look forward to receiving your revised manuscript.

Kind regards,

Kohei Kajiwara

Academic Editor

PLOS One

2. Please amend your manuscript to include your abstract after the title page.

3. Please include a caption for figure 1.

Additional Editor Comments:

Thank you for your submission.

This study, which presents a scoping review on the timing of advance care planning (ACP), addresses an important focus. However, as noted by the reviewers, further clarification and additional details are required regarding the research methods and the presentation of the results. In light of these points, I also believe that the discussion section needs to be expanded.

Please reconsider and revise the manuscript accordingly in response to the reviewers’ comments.

Reviewers' comments:

Reviewer's Responses to Questions

**Comments to the Author**

1. Is the manuscript technically sound, and do the data support the conclusions?

Reviewer #1: Partly

Reviewer #2: Yes

2. Has the statistical analysis been performed appropriately and rigorously?

Reviewer #1: N/A

Reviewer #2: Yes

3. Have the authors made all data underlying the findings in their manuscript fully available?

Reviewer #1: No

Reviewer #2: Yes

4. Is the manuscript presented in an intelligible fashion and written in standard English?

Reviewer #1: Yes

Reviewer #2: Yes

Reviewer #1: Review Comments

Thank you for the opportunity to review this manuscript. The topic is highly relevant, as the importance of ACP is increasing globally. I particularly appreciate that the analysis considered not only patients’ perspectives but also those of their caregivers, which adds significant value to the study. While some limitations are acknowledged, I believe this research provides a solid foundation for future studies to advance the field.

To further strengthen the manuscript, I kindly suggest considering the following points for clarification and improvement:

Major Comments

1) Study Selection Could you please clarify whether the screening method was conducted in accordance with established guidelines?

After reviewing both the full text and the protocol, it was not entirely clear whether double screening was applied to all documents during the screening process. For example:

•Protocol:“In phase 2, CB will screen the remaining articles for full-text eligibility. EW, SB, and LN will each screen a substantial part of full texts for eligibility.”

•Manuscript:“CB screened 8867 records for eligibility based on title and abstract, with 854 (9.6%) also independently screened by EW, LN, or SB. This led to 84 articles selected for full-text review. CB assessed all full texts, with 15 (17.9%) additionally reviewed independently by EW, LN, or SB, resulting in 25 articles meeting the inclusion criteria.The updated search up to May 2025 retrieved 2816 additional records, including 518 duplicates. CB screened the remaining 2298 records, selecting 22 for full-text review, of which four met the inclusion criteria.”

If the screening process adhered to the guidelines, please make this explicit in the manuscript to avoid any ambiguity.

2) Table 3 – “Uncertainty” subtheme

Regarding the subtheme “Uncertainty” under professional-related challenges related to the timing of ACP: even after reviewing the code tree, the rationale for setting this subtheme was unclear. It is not mentioned in the Results section, although the Discussion addresses it in detail.

Providing a more detailed description of the results related to this subtheme would enhance the clarity and persuasiveness of your findings.

Minor Comments

1.S1 Search String

Please use “PubMed” consistently in the main text.

2.Figure 1

The notation “n=11.165” and “n=11.059” uses a period (.) instead of a comma (,).

A comma would be preferable for consistency. Additionally, this notation appears only in these items, creating inconsistency. Standardizing the format would improve readability.

3.Table 1

Consider organizing the order (e.g., by year) to make the research overview easier to grasp.

4.Table 2 & S2 (Code Tree)

Including citation symbols to indicate the source literature for each data point would be helpful. Currently, it is difficult to identify which literature the data in Table 2 originates from. Adding these references would also strengthen the credibility of the data.

I hope these suggestions are helpful in refining your manuscript. Thank you again for your valuable contribution to this important area of research.

Reviewer #2: Thank you for the opportunity to review this paper. The authors conducted a scoping review on the "appropriate timing of discussions," an important issue for expanding advance care planning. I found this scoping review interesting. I would like to offer several comments to improve this manuscript.

1. A scoping review is a method for comprehensively mapping existing literature, but this study focused only on two databases: PubMed and CINAHL. While I understand that experienced librarians were involved, I would like to ask you to explain why these two search engines were selected.

2. Is it correct to assume that all articles were independently screened using "CB" and "EW, LN, SB"?

3. It is expected that results will differ depending on the target disease (cancer, heart disease, lung disease). Please clarify why you chose not to address differences by disease.

4. As the authors state, this study integrates studies from widely different cultural backgrounds, healthcare systems, and definitions of ACP, so caution is required when generalizing themes. In particular, patients' and families' perceptions of "appropriate timing" may differ significantly between countries with well-established ACP systems (e.g., the Netherlands, the UK) and those without (e.g., Brazil, Asian countries). Therefore, please consider further how national, cultural, and institutional differences may affect the results.

5. A strength of this study is that it addressed the perspectives of both patients and families, but the results do not clearly show the differences between the two. Consider indicating the differences between the patient and family perspectives.

6. The detailed structure of Table 1 is acceptable, but it would be easier to read if it were classified by disease.

7. Because ACP was handled differently in each study, it would be desirable to provide a brief summary at the beginning.

**Do you want your identity to be public for this peer review?** For information about this choice, including consent withdrawal, please see our Privacy Policy

Reviewer #1: No

Reviewer #2: No

You may also use PLOS’s free figure tool, NAAS, to help you prepare publication quality figures: https://journals.plos.org/plosone/s/figures#loc-tools-for-figure-preparation

---

## [Author Response · Author response to Decision Letter 1]

26 Jan 2026

Dear Dr. Kohei Kajiwara and distinguished reviewers,

Thank you for your careful and thorough reviews. We greatly appreciate the reviewers’ thoughtful appraisal of our manuscript and the opportunity to revise it. We have carefully considered all comments and revised the manuscript accordingly. In addition to addressing the reviewers’ points, we have also ensured that the manuscript complies with the journal’s formatting requirements and incorporated additional clarifications suggested by the editor, which largely overlap with the reviewers’ recommendations.

Below, we provide the reviewers’ comments (Comment), our responses to these comments and questions (Response, in italics), and the corresponding changes made in the manuscript (Revisions, in bold).

Your valuable feedback has helped us to strengthen the manuscript. Thank you for your consideration, we look forward to your response.

Kind regards, also on behalf of the co-authors,

Carolien Burghout

---

## [Decision Letter · Decision Letter 1]

20 Feb 2026

PONE-D-25-49791R1What is the appropriate timing for advance care planning according to patients and their relatives? A scoping review .PLOS One?

Dear Dr. <!--StartFragmentBurghout<!--EndFragment,

Thank you for submitting your manuscript to PLOS ONE. After careful consideration, we feel that it has merit but does not fully meet PLOS ONE’s publication criteria as it currently stands. Therefore, we invite you to submit a revised version of the manuscript that addresses the points raised during the review process.

We look forward to receiving your revised manuscript.

Kind regards,

Kohei Kajiwara

Academic Editor

PLOS One

Journal Requirements:

Additional Editor Comments:

Thank you for your submission. The reviewers’ comments have been appropriately addressed overall; however, some minor points still require revision. I therefore invite you to revise the manuscript accordingly and resubmit the revised version.

Reviewers' comments:

Reviewer's Responses to Questions

**Comments to the Author**

Reviewer #1: All comments have been addressed

Reviewer #2: All comments have been addressed

2. Is the manuscript technically sound, and do the data support the conclusions?

Reviewer #1: Partly

Reviewer #2: Yes

3. Has the statistical analysis been performed appropriately and rigorously?

Reviewer #1: N/A

Reviewer #2: Yes

4. Have the authors made all data underlying the findings in their manuscript fully available?

Reviewer #1: Yes

Reviewer #2: Yes

5. Is the manuscript presented in an intelligible fashion and written in standard English?

Reviewer #1: Yes

Reviewer #2: Yes

Reviewer #1: Thank you for clarifying the study selection procedure and for adding detailed descriptions in both the Methods and Results sections. The transparency of the screening process has clearly improved.

Since you now state that the procedure was conducted “according to international guidelines,” I would encourage you to specify which guideline or methodological source you are referring to.

If your approach is based on resource-efficient screening methods—such as those described by Garritty et al. (2022)—it would be helpful to cite this work directly and briefly explain that partial double-screening is an accepted method within this framework.

Without this clarification, readers may assume adherence to more traditional guidelines (e.g., PRISMA or JBI), which generally recommend independent screening by two reviewers for all records.

Therefore, explicitly naming and citing the guideline or methodological paper that supports your approach would prevent misunderstanding and strengthen the credibility of the methods section.

Overall, your revisions move the manuscript in a positive direction, and further clarification on this point will help ensure complete methodological transparency.

You may consider adding the following clarification to avoid any misunderstanding regarding your screening methodology:

“Our screening approach followed resource‑efficient methods as described by Garritty et al. (2022), which support single‑reviewer screening with partial independent verification. This method maintains high accuracy while reducing workload.”

This wording clearly indicates the methodological basis for your screening process and helps ensure that readers do not assume adherence to more traditional full double‑screening models.

Reviewer #2: The authors have provided clear, thoughtful, and comprehensive responses to all reviewer comments. The revisions substantially improve the clarity, structure, and contextual depth of the manuscript.

Overall, the responses are satisfactory, and the revisions meaningfully address the concerns raised. The manuscript is notably improved and is closer to being suitable for publication.

**Do you want your identity to be public for this peer review?** For information about this choice, including consent withdrawal, please see our Privacy Policy

Reviewer #1: No

Reviewer #2: No

---

## [Author Response · Author response to Decision Letter 2]

27 Feb 2026

Dear distinguished reviewers,

Thank you for your careful evaluation of our revised manuscript and for the opportunity to address the remaining minor point. We greatly appreciate the reviewers’ thoughtful and constructive feedback, which has helped us to further clarify the method of study selection.

We provide the reviewers’ comment (Comment), our response (Response, in italics), and the corresponding change made in the manuscript (Revision, in bold).

We believe the manuscript has been further strengthened and is now fully aligned with the reviewers’ and editor’s recommendations. We look forward to your response.

Kind regards, also on behalf of the co-authors,

Carolien Burghout

---

## [Editor Report · Decision Letter 2]

3 Mar 2026

What is the appropriate timing for advance care planning according to patients and their relatives? A scoping review .

PONE-D-25-49791R2

Dear Dr. Carolien Burghout,

We’re pleased to inform you that your manuscript has been judged scientifically suitable for publication and will be formally accepted for publication once it meets all outstanding technical requirements.

Kind regards,

Kohei Kajiwara

Academic Editor

PLOS One

Additional Editor Comments:

Thank you for submitting the revised manuscript. I believe that the manuscript has been appropriately revised in accordance with the comments.